# Partial Identification with Noisy Covariates:
# A Robust Optimization Approach

**Wenshuo Guo**                       WGUO@CS.BERKELEY.EDU
*University of California, Berkeley*

**Mingzhang Yin**                       MY2674@COLUMBIA.EDU
*Columbia University*

**Yixin Wang**                        YIXINW@UMICH.EDU
*University of Michigan*

**Michael I. Jordan**                    JORDAN@CS.BERKELEY.EDU
*University of California, Berkeley*

**Editors:** Bernhard Scholkopf, Caroline Uhler and Kun Zhang

## Abstract

Causal inference from observational datasets often relies on measuring and adjusting for covariates. In practice, measurements of the covariates can often be noisy and/or biased, or only measurements of their proxies may be available. Directly adjusting for these imperfect measurements of the covariates can lead to biased causal estimates. Moreover, without additional assumptions, the causal effects are not point-identifiable due to the noise in these measurements. To this end, we study the partial identification of causal effects given noisy covariates, under a user-specified assumption on the noise level. The key observation is that we can formulate the identification of the average treatment effects (ATE) as a robust optimization problem. This formulation leads to an efficient robust optimization algorithm that bounds the ATE with noisy covariates. We show that this robust optimization approach can extend a wide range of causal adjustment methods to perform partial identification, including backdoor adjustment, inverse propensity score weighting, double machine learning, and front door adjustment. Across synthetic and real datasets, we find that this approach provides ATE bounds with a higher coverage probability than existing methods.

**Keywords:** Causal inference; Robust optimization; Noisy covariates

## 1. Introduction

Estimating the causal effect of an intervention is a problem that arises in countless domains, with examples including identifying the effect of medical treatments (Connors et al., 1996), evaluating the effectiveness of recommender systems (Schnabel et al., 2016; Wang et al., 2020), and assessing the impact of educational methods (Gustafsson, 2013). In many of these settings, the challenge is to identify causal effects from observational data, and a core problem is that naive inference can be biased by *confounders*, which are variables that affect both the intervention and the outcomes. For example, in identifying the effect of college education on earnings for students, the scholastic ability is a confounder (Card, 1999)—it can affect both whether the student can be admitted to a college and how much he/she may earn after graduation. As a result, the observed increase in earnings associated with attending college is confounded by the effect of the scholastic ability and thus cannot accurately represent the causal effect of college education.

A common approach to addressing confounding bias is aiming to measure all of the confounders and adjust for them (Imbens and Rubin, 2015). In practice, however, measurements of the confounders can often be noisy or biased. Moreover, sometimes we only have access to proxies of the confounders. For instance, in the example of college education and earnings, confounders are often measured via surveys and thus are typically biased and incomplete—participants may not be willing to discuss aspects of their family backgrounds or reveal their access to alternative educational or career options. Some confounders are difficult to measure by definition—for example, students' innate cognitive abilities—and in such cases we generally only have access to proxies.

While noisy covariates (including both noisy measurements and confounder proxies) are generally needed to perform causal inference, directly adjusting for such covariates can lead to biased causal estimates (Fuller, 2009). Moreover, it is well known that, without further assumptions on the causal model, the causal effects are not point-identifiable given only noisy covariates (Carroll et al., 2006; Schennach, 2016; Ogburn and Vanderweele, 2013; Lockwood and McCaffrey, 2016). In other words, with access to only noisy covariates, it may be impossible to pinpoint the causal effect of interest even with infinite data. How then can noisy covariates inform causal inference?

In this paper, we leverage the noisy covariates to perform partial identification of the causal effects. Given a user-specified assumption on the noise level, we develop an algorithm for partial identification using robust optimization. This approach capitalizes on two observations: (1) the causal effects of interest are identifiable given the (unobserved) true joint distribution of treatments, outcomes, and all (noiseless) covariates; (2) the dataset with noisy covariates places constraints on what this joint distribution can be. These observations allow us to turn the task of partial identification into a robust optimization problem.

In more detail, we formulate partial identification as the following robust optimization problem. We first consider an uncertainty set of all possible underlying joint distribution of treatments, outcomes, and the noiseless covariates subject to the constraints. Then we find the maximum (or minimum) possible causal effects that a distribution in this set can plausibly lead to. This approach propagates the uncertainty in the true data distribution (due to covariate noise) downstream to the uncertainty in the causal estimation, leading to partial identification intervals of the causal effects.

Taking this optimization perspective on partial identification, we develop an algorithm that efficiently solves the robust optimization problem and computes the bounds on the causal effect of interest. This algorithm is applicable to a wide variety of causal adjustment methods, including backdoor adjustment, frontdoor adjustment, inverse propensity score weighting (IPW), and double machine learning, which we demonstrate. Across simulated and real datasets, we find that this approach can produce tight bounds on causal effects that cover the true average treatment effect.

**Contributions.** We propose a robust optimization approach to partial identification given noisy covariates. The key idea is to formulate partial identification with noisy covariates as a robust optimization problem. We provide an efficient algorithm to solve this robust optimization program, thereby obtaining upper and lower bounds on the causal effects. We demonstrate the general applicability of this approach by applying it to a variety of causal adjustment methods. Finally, we demonstrate the effectiveness of the approach across empirical studies with synthetic and real data.

**Related work.** This work draws on several threads of research in measurement noise, proxy variables, and robust optimization.

The first is on measurement noise and proxy variables in causal inference. This subject has a long history in the literature (Wickens, 1972; Frost, 1979), where there have been a variety of pro-

posals for recovering causal effects either heuristically or with additional model assumptions (Carroll et al., 2006; Schennach, 2016; Ogburn and Vanderweele, 2013; Lockwood and McCaffrey, 2016). Recent examples include Louizos et al. (2017), who use variational autoencoders as a heuristic way to recover the latent confounders; and Kallus et al. (2018), who use matrix factorization to infer the confounders from the noisy covariates assuming that the data-generating process follows a linear outcome model. Given proxy variables of unmeasured confounders, Kuroki and Pearl (2014) and Miao et al. (2018) propose specific technical conditions under which causal effects can be restored. These results have been extended to a variety of other settings (Tchetgen et al., 2020; Shi et al., 2020; Cui et al., 2020; Shpitser et al., 2021; Dukes et al., 2021; Ying et al., 2021; Shi et al., 2021). Finally, Imai and Yamamoto (2010) seek to partially identify ATE under measurement error using constrained linear optimization. More recently, Finkelstein et al. (2020); Duarte et al. (2021); Zhang et al. (2021); Zhang and Bareinboim (2021b); Balke and Pearl (1994, 1997); Ramsahai and Spirtes (2012); Bonet (2013); Heckman and Vytlacil (2001); Sachs et al. (2020); Geiger and Meek (1999) develop optimization formulations for partial identification. Most of this work focuses on discrete variables under settings including unobserved confounding and/or measurement error. Our work is complementary to this work; we focus on noisy covariates but can handle certain settings with continuous variables, without relying on additional compliance assumptions.

Noisy covariates or proxy variables are not generally sufficient to identify causal effects as they violate the "no unobserved confounders" assumption. Therefore, handling noisy covariates relates to sensitivity analyses that seek partial identification; i.e., bounds on the average treatment effect (ATE) (Liu et al., 2013; Richardson et al., 2014; Imbens, 2003; Veitch and Zaveri, 2020; Dorie et al., 2016; Cinelli and Hazlett, 2020; Cinelli et al., 2019; Franks et al., 2019; Shen et al., 2011; Hsu and Small, 2013; Bonvini and Kennedy, 2020; Rosenbaum et al., 2010; Zhao et al., 2017; Yadlowsky et al., 2018; Zhang and Bareinboim, 2021a; Yin et al., 2021). In this vein, the work of Yadlowsky et al. (2018) is most related. They propose a loss-minimization approach that quantifies bounds on the conditional average treatment effect (CATE). Their approach requires the unobserved confounder to satisfy a constraint that bounds the effect on the odds of treatment selection.

The second thread of related work concerns robust optimization, which is a core ingredient of the robust causal inference approach we develop. We adopt a minimax formulation of a two-player game where the uncertainty is adversarial, and one minimizes a worst-case objective over a feasible set (Ben-Tal et al., 2009; Bertsimas et al., 2011). For example, the noise may be contained in a unit-norm ball around the input data. To solve the robust optimization problem, we build on a recent line of work on distributionally robust optimization (DRO) which assumes that the uncertain distributions underlying the data have support within a certain set (Namkoong and Duchi, 2016; Duchi and Namkoong, 2018; Li et al., 2019).

## 2. Preliminaries: Potential Outcomes, ATE Estimation, and Noisy Covariates

In this section, we set up the causal inference problem with noisy covariates and formalize the assumptions required by the partial identification of ATE.

**Potential outcome and ATE estimation.** Let $D^* = (X, Y, Z)$ denote a dataset, where $X$ represents a vector of (possibly unobserved) noiseless covariates. Denote $Z$ as a binary treatment random variable, with $0$ and $1$ being the labels for control and active treatments, respectively. Further, let $Y$ denote the outcome. We use the potential outcomes notation to define causal quantities (Neyman, 1923; Rubin, 1974): For each realization of the level of treatment, $z \in \{0, 1\}$, we assume that there

exists a potential outcome, $Y(z)$, representing the outcome had the subject been given treatment $z$ (possibly contrary to fact). Then, the observed outcome is $Y = Y(Z) = ZY(1) + (1 - Z)Y(0)$. We focus on estimating the average treatment effect (ATE):

$$\tau = \mathbb{E}[Y(1) - Y(0)]. \tag{1}$$

To focus on the causal inference challenge due to noisy covariates, we assume that the ATE is identifiable with the (potentially unobserved) noiseless data.

**Assumption 1 (Identifiability of ATE given noiseless covariates)** *The ATE is identifiable ([Pearl, 1995](#)) given the (unobserved) noiseless covariates, in addition to the observed treatment and outcome, namely the ATE can be written as a functional of the joint distribution of $D^* = (X, Y, Z)$.*

Assumption 1 ensures the identifiability of ATE if we had access to the noiseless covariates. This assumption is satisfied when the noiseless covariates $X$ meet identification conditions for ATE such as the backdoor criterion, the positivity condition and weak unconfoundedness ([Rosenbaum and Rubin](#), 1983), and the frontdoor criterion ([Pearl](#), 2009).

**Noisy covariates.** Though the noiseless data $D^* = (X, Y, Z)$ can identify the ATE, we often do not have access to such a dataset. Instead, we only have access to a dataset with noisy covariates $\tilde{X}$. We denote the observed dataset as $D = (\tilde{X}, Y, Z)$.

The noisy covariates $\tilde{X}$ shall potentially still provide information about $X$ despite the noise. To describe to what extent can $\tilde{X}$ inform $X$, we rely on an assumption about the noise level.

**Assumption 2 (Noise level)** *The TV distance between the distributions of $X$ and $\tilde{X}$, conditional on the treatment variable $Z$, is bounded by a constant $\gamma_z$:*

$$\mathrm{TV}(p_{x|z}, p_{\tilde{x}|z}) \leq \gamma_z, z \in \{0, 1\}, \tag{2}$$

*where $p_{x|z}$ is the distribution of the unobserved noiseless covariates for the treatment or control group, $X|Z = z \sim p_{x|z}$, $p_{\tilde{x}|z}$ is the distribution of the noisy covariates $\tilde{X}$, $\tilde{X}|Z = z \sim p_{\tilde{x}|z}$, and the TV distance between two probability distributions $p$ and $q$ is defined as follows:*

$$\mathrm{TV}(p, q) = \inf_{\pi} \mathbb{E}_{X,Y \sim \pi(x,y)}[\mathbb{1}(x \neq y)] \tag{3}$$

$$\text{s.t.} \int \pi(x, y) dy = p(x), \int \pi(x, y) dx = q(y),$$

*where $\pi$ represents a coupling between $p$ and $q$ (cf. [Villani, 2008](#)).*

Assumption 2 provides a convenient way to characterize how far away the noisy covariates $\tilde{X}$ are from their (unobserved) noiseless counterpart $X$, where the further away the noisy covariates $\tilde{X}$ is from $X$, the less informative $\tilde{X}$ is for causal inference. The upper bound of the TV distance $\gamma_z$ in Assumption 2 is a user-specified parameter, which is often specified by domain experts or estimated from auxiliary datasets. In particular, the TV distance as a distance to quantify the noise level in Assumption 2 because of the computational tractability of TV distance in robust optimization, which we will detail in § 3 and appendix A.

Many existing noise models imply the TV bound on distributions in Assumption 2. We illustrate a few of these models below.

1. *Huber contamination model.* Suppose the noisy covariates deviate from the noiseless covariates following the Huber contamination model

$$p_{\tilde{x}|z} = (1 - \gamma_z)p_{x|z} + \gamma_z h_{\tilde{x}|z},$$

where $h_{\tilde{x}|z}$ can be any arbitrary distribution. Then the noisy covariates satisfy $\mathrm{TV}(p_{x|z}, p_{\tilde{x}|z}) \leq \gamma_z$.

2. *Misclassification error.* Suppose the noisy covariates $\tilde{X}$ are discrete and their misclassification error satisfies

$$\max_x \left| P(X = x|Z = z) - P(\tilde{X} = x|Z = z) \right| < \gamma_z, \quad z \in \{0, 1\}.$$

Then we also have $\mathrm{TV}(p_{x|z}, p_{\tilde{x}|z}) \leq \gamma_z$.

3. *Exponential tilting model.* Suppose the distribution of $\tilde{X}$ is an exponential tilted version of $X$, and both distributions belong to the exponential family,

$$p(x|z) = \exp(\eta(\theta_z) \cdot T(x) + A(\theta_z) + B(x)), \qquad p(\tilde{x}|z) = \exp(\eta(\tilde{\theta}_z) \cdot T(\tilde{x}) + A(\tilde{\theta}_z) + B(\tilde{x})).$$

Then $\mathrm{TV}(p_{x|z}, p_{\tilde{x}|z}) \leq \gamma_z \triangleq \sqrt{\frac{1}{2}D_A(\tilde{\theta}_z, \theta_z)}$, where $D_A(\cdot)$ is the Bregman divergence with function $A$ (Banerjee et al., 2005).

4. *Other models.* For general noise models for $\tilde{X}$, one can approximately estimate the TV bound $\gamma_z$ in Assumption 2 by drawing samples from $p_{\tilde{x}|z}$ and $p_{x|z}$ respectively, calculating the KL divergence estimates (Pérez-Cruz, 2008; Belghazi et al., 2018), and applying Pinsker's inequality.

Finally, Assumption 2 specifies a noise-level assumption on the conditional distributions $p_{x|z}$ and $p_{\tilde{x}|z}$. One can similarly specify the noise level for the marginal distributions of the covariates, $p_x, p_{\tilde{x}}$, or the conditional distributions given both the treatments and the outcomes, $p_{x|y,z}, p_{\tilde{x}|y,z}$. Here we consider the use of $p_{x|z}$ and $p_{\tilde{x}|z}$ as a demonstrative example, formulating the partial identification task into a robust optimization problem in § 3. Similar derivations apply to versions of Assumption 2 with other distributions.

## 3. Partial Identification with Noisy Covariates

Though Assumption 1 ensures that the ATE is identifiable given the noiseless covariates $X$, the ATE is not point-identifiable without further assumptions given only the noisy covariates $\tilde{X}$—there exist many values of ATE that are all compatible with the observed distribution of $D$ (Carroll et al., 2006; Schennach, 2016; Ogburn and Vanderweele, 2013; Lockwood and McCaffrey, 2016).

Given this lack of point identifiability, we focus on partial identification of ATE. Instead of providing a point estimate of the ATE, we aim to bound the ATE given the dataset with noisy covariates $D = (\tilde{X}, Y, Z)$. In particular, we develop an optimization approach to partial identification. The key idea is to cast the task of partial identification as a robust optimization problem. We consider the set of all joint distributions of the (unobserved) noiseless data $P(X, Y, Z)$ that are compatible with the observed noisy data $P(\tilde{X}, Y, Z)$ under the noise-level assumption (Assumption 2). Then the minimum and maximum value of ATE resulting from these joint distributions shall bound the ATE. It turns out that finding the minimum and maximum can be turned into a robust optimization problem, for which we develop an algorithm to solve.

In the rest of this section, we begin with the parametric approach to estimate ATE given noiseless data $D^* = (X, Y, Z)$; it can be written as an optimization problem as the parameter model is fitted via maximum likelihood. We then expand this optimization problem to consider the dataset with noisy covariates $D = (\tilde{X}, Y, Z)$, which results in a robust optimization problem. We will derive an efficient algorithm to solve the optimization and demonstrate its general applicability to common causal adjustment methods in § 4.

### 3.1. The parametric modeling approach to ATE estimation

We begin with estimating the ATE assuming oracle access to noiseless data $D^* = (X, Y, Z)$.[1] The ATE is identifiable given $D^*$ due to Assumption 1.

We adopt a parametric modeling approach to ATE estimation, where we posit a parametric model for the joint distribution $p_{x,y,z}$ or its components required by the identification formula. Specifically, we first posit a parametric model for the joint distribution or its relevant conditionals. For example, we may posit that the joint distribution $p_{x,y,z}$ follows a parametric model $\{p_\theta(x, y, z) : \theta \in \Theta\}$, where $\Theta$ is the parameter space. As another example, one may posit a parametric model only for a conditional component of $p_{x,y,z}$, e.g. $p_\theta(x, y, z) = p_x \times p_\theta(z|x) \times p_{y|x,z}$, where the conditional $p_{z|x}$ follows a statistical model parameterized by $\theta$, $\{p_\theta(z|x) : \theta \in \Theta\}$. Given the parametric model, we find the likelihood maximizing parameter $\theta$

$$\hat{\theta} = \arg\max_\theta L_n(\theta \, ; \, p_{x,y,z}), \tag{4}$$

where $L_n(\theta \, ; \, p_{x,y,z}) \triangleq \mathbb{E}_{p_{x,y,z}}[\log p_\theta(x, y, z)]$ is the likelihood of the data $D^* = (X, Y, Z)$ at parameter $\theta$. Finally, we plug in the fitted parametric model for causal estimation

$$\hat{\tau} = Q(p_{\hat{\theta}}(x, y, z)), \tag{5}$$

where $p_{\hat{\theta}}(x, y, z)$ is the joint distribution of $(X, Y, Z)$ implied by the posited statistical model at the optimal parameter $\hat{\theta}$, and $Q(\cdot)$ is the causal identification functional mapping the joint distribution of $(X, Y, Z)$ to the ATE $\tau$.

As an example, suppose we adopt the backdoor adjustment for estimation. We first posit a parametric model for $p_{y|x,z}$ with density $p_\theta(y|x, z) = \mathcal{N}(f(x, z; \theta), 1^2)$, find the maximum likelihood parameters $\hat{\theta}$ by maximizing $L_n(\theta; D^*)$, the Gaussian likelihood of the $n$ data points in $D^*$ given parameter $\theta$, and finally calculate the ATE estimate following the backdoor adjustment $\hat{\tau} = \mathbb{E}_X[f(X, 1; \hat{\theta})] - \mathbb{E}_X[f(X, 0; \hat{\theta})]$.

### 3.2. Partial identification as robust optimization

The parametric approach to ATE estimation relies on having access to noiseless covariates $X$. However, we often only have access to the dataset with noisy covariates $D = (\tilde{X}, Y, Z)$, and the ATE is no longer point identifiable; they may only partially identify the ATE. Then how can we extend the parametric approach to partially identify the ATE?

**Partial identification of ATE as a robust optimization.** To perform partial identification, we extend the optimization problem of Eq. 4 to a robust optimization. The key observation is that, though the noiseless data distribution $p_{x,y,z}$ is unobserved, the observed noisy data distribution

---

1. The dataset contains $n$ i.i.d. data points $\{X_i, Y_i, Z_i\}_{i=1}^n$. We suppress the data index for notation simplicity.

$p_{\tilde{x},y,z}$, together with the noise-level assumption (Assumption 2), characterizes an uncertainty set of $p_{x,y,z}$, which further leads to an uncertainty set of ATE, following the same identification formula in Eq. 5. If the uncertainty set of $p_{x,y,z}$ contains the true $p_{x,y,z}$, then its resulting uncertainty set for the ATE shall also contain the true ATE. In other words, the maximum and minimum of this ATE uncertainty set bound the true ATE, hence partial identification.

Formally, we obtain the partial identification interval by solving the following optimization problem analogous to the one in the parametric approach (Eqs. 4 and 5). Denote the uncertainty set of $p_{x,y,z}$ as $\mathcal{P}_{X,Y,Z}$. Then the lower bound of ATE $\hat{\tau}_L$ is obtained by

$$\hat{\tau}_L = \min_{p_{x,y,z} \in \mathcal{P}_{X,Y,Z}} Q(p_{\hat{\theta}}(x, y, z)) \tag{6}$$

$$\text{s.t.} \quad \hat{\theta} = \arg\max L_n(\theta \,;\, p_{x,y,z}), \tag{7}$$

which is a form of a distributionally robust optimization (DRO). We can similarly obtain the upper bound $\hat{\tau}_U$ by replacing $\min$ with $\max$, and the partial identification interval estimate for the ATE $\tau$ is $[\hat{\tau}_L, \hat{\tau}_U]$. It is similar to Eqs. 4 and 5: the parametric model is similarly placed on the noiseless data $p_{x,y,z}$. The only difference is that $p_{x,y,z}$ is unobserved; we have to calculate Eqs. 4 and 5 for all possible $p_{x,y,z}$ within the uncertainty set $\mathcal{P}_{X,Y,Z}$. This formulation of partial identification as robust optimization produces tight partial identification bounds. The tightness is achieved by construction, as any $p_{x,y,z}$ that achieves the minimum and maximum value of the objective is compatible with the observed data and the posited statistical model due to the constraint of $p_{x,y,z} \in \mathcal{P}_{X,Y,Z}$. Below we discuss some practical aspects of partial identification: constructing the uncertainty set, solving the robust optimization problem, and statistical inference of the partial identification bounds.

**The uncertainty set $\mathcal{P}_{X,Y,Z}$.** To construct the uncertainty set of $p_{x,y,z}$, we focus on characterizing $p_{x|y,z}$, the conditional distribution of the noiseless covariates $X$ given $Y, Z$. The reason is that the conditional distribution $p_{x|y,z}$, along with treatment and outcome distribution $p_{y,z}$, fully determines the joint $p_{x,y,z} = p_{x|y,z} \times p_{y,z}$. Thus the ATE can be identified by Eq. 5 under Assumption 1.

To construct the uncertainty set for $p_{x|y,z}$, we resort to Assumption 2, which requires that $p_{x|z} \in \{\bar{p}_{x|z} : \text{TV}(\bar{p}_{x|z}, p_{\tilde{x}|z}) \leq \gamma_z\}$ for $z \in \{0, 1\}$. Let us denote $\boldsymbol{\gamma} = (\gamma_z)_{z \in \{0,1\}}$. Thus, by the chain rule, the uncertainty set of $p_{x,y,z}$ is

$$\mathcal{P}_{X,Y,Z}(p_{\tilde{x},y,z} \,;\, \boldsymbol{\gamma}) = \left\{ p_{y,z} \times \bar{p}_{x|y,z} : \text{TV}\left( \int \bar{p}_{x|y,z} \times p_{y|z} dy, p_{\tilde{x}|z} \right) \leq \gamma_z, \forall y \in \mathcal{Y}, z \in \{0, 1\} \right\}.$$

**Solving the robust optimization problem.** Eqs. 6 and 7 define a distributionally robust optimization (DRO) problem, which generally takes the form of a minimax optimization, $\min_{\theta \in \Theta} \max_{q:D(q,p) \leq \gamma} \mathbb{E}_{X,Y \sim q}[l(\theta, X, Y)]$, where $D$ is some divergence metric between the distributions $p$ and $q$, and $l : \Theta \times \mathcal{X} \times \mathcal{Y} \to \mathbb{R}$ (Duchi and Namkoong, 2018). Such problems can be solved by re-writing Eqs. 6 and 7 using via a Lagrangian formulation:

$$\hat{\tau}_L = \min_{\theta} \min_{\substack{p_{x,y,z} \in \\ \mathcal{P}_{X,Y,Z}(p_{\tilde{x},y,z} \,;\, \boldsymbol{\gamma})}} \max_{\lambda \geq 0} \quad Q(p_{\theta}(x, y, z)) - \lambda \cdot L_n(\theta \,;\, p_{x,y,z}), \tag{8}$$

when the function $L_n(\cdot)$ is upper bounded. We can then apply existing methods that efficiently and optimally solve the DRO problem different divergence metrics $D$ (Namkoong and Duchi, 2016; Li et al., 2019; Esfahani and Kuhn., 2018), equipped with finite-sample convergence rates analyzed in Duchi and Namkoong (2018). (See Appendix A for details.)

**Statistical inference of the partial identification bounds.** The robust optimization problem (Eqs. 6 and 7) produces point estimates for the partial identification bounds of ATE. To assess the sampling uncertainty of these bounds, one can invoke standard statistical inference tools (Duchi and Namkoong, 2018). Specifically, we consider a separate TV ball around the observed data distribution $\{\bar{p}_{\tilde{x},y,z} : \mathrm{TV}(\bar{p}_{\tilde{x},y,z}, p_{\tilde{x},y,z}) \le \rho/n\}$, where $n$ is the sample size and $\rho = \chi^2_{1,1-\alpha}$ is the $(1-\alpha)$-quantile of the $\chi^2_1$ distribution. We can then obtain upper and lower confidence limits for $\hat{\tau}_L$ and $\hat{\tau}_U$. For instance, for the lower bound of ATE $\hat{\tau}_L$, its upper and lower confidence limits are

$$u_{\hat{\tau}_L} = \min_{\theta} \min_{\substack{p_{x,y,z} \in \\ \mathcal{P}_{X,Y,Z}(\bar{p}_{\tilde{x},y,z}\,;\boldsymbol{\gamma})}} \max_{\substack{\mathrm{TV}(\bar{p}_{\tilde{x},y,z}, p_{\tilde{x},y,z}) \\ \le \rho/n}} \max_{\lambda \ge 0} \quad Q(p_{\hat{\theta}}(x,y,z)) - \lambda \cdot L_n(\theta\,;p_{x,y,z}),$$

$$l_{\hat{\tau}_L} = \min_{\theta} \min_{\substack{p_{x,y,z} \in \\ \mathcal{P}_{X,Y,Z}(\bar{p}_{\tilde{x},y,z}\,;\boldsymbol{\gamma})}} \min_{\substack{\mathrm{TV}(\bar{p}_{\tilde{x},y,z}, p_{\tilde{x},y,z}) \\ \le \rho/n}} \max_{\lambda \ge 0} \quad Q(p_{\hat{\theta}}(x,y,z)) - \lambda \cdot L_n(\theta\,;p_{x,y,z}).$$

Similarly, one can obtain the upper and low confidence limits of the upper bound $\hat{\tau}_U$. Importantly, these confidence limits $[l_{\hat{\tau}_L}, u_{\hat{\tau}_L}]$ quantify the sampling uncertainty of $\hat{\tau}_L$ because we do not have access to the true distribution $p_{\tilde{x},y,z}$. In contrast, the partial identification bounds $[\hat{\tau}_L, \hat{\tau}_U]$ quantify the identification uncertainty of $\tau$ due to noisy covariates. As the sample size $n$ increases, the confidence intervals $[l_{\hat{\tau}_L}, u_{\hat{\tau}_L}]$ and $[l_{\hat{\tau}_U}, u_{\hat{\tau}_U}]$ shrink to a point mass, but the identification interval $[\hat{\tau}_L, \hat{\tau}_U]$ does not shrink.

In more detail, suppose the propensity score given all observed covariates is $e(X_{\mathrm{inc}}) \triangleq P(Z = 1 \,|\, X_{\mathrm{inc}})$, where $X_{\mathrm{inc}}$ denotes all the observed covariates, which may not include all confounders and satisfy weak unconfoundedness (Imbens and Rubin, 2015). Further denote the propensity score given all confounders $e(X_{\mathrm{full}}) \triangleq P(Z = 1 \,|\, X_{\mathrm{full}})$, where $X_{\mathrm{full}}$ satisfy weak unconfoundedness $Z \perp Y(1), Y(0) \,|\, X_{\mathrm{full}}$. Then the robust optimization approach to partial identification can be used to obtain ATE bounds under the sensitivity assumptions like $\mathrm{TV}(p_{e(x_{\mathrm{inc}})}, p_{e(x_{\mathrm{full}})}) \le \gamma$ or $\mathrm{TV}(p_{e(x_{\mathrm{inc}})|z}, p_{e(x_{\mathrm{full}})|z}) \le \gamma_z$ for some constants $\gamma_z, \gamma$.

## 4. Applications to Common Causal Adjustment Methods

In this section, we apply the general robust optimization strategy to a variety of popular causal adjustment methods. In particular, we instantiate the estimator (Eq. 6) for three adjustment methods, backdoor adjustment, inverse propensity weighting (IPW), and frontdoor adjustment. (We further demonstrate the application to double machine learning in Appendix B.) For each adjustment method, we first provide a brief review of the standard procedure, then demonstrate how it can be augmented to perform partial identification given noisy covariates. Specifically, we write the objective $Q(\cdot)$ and the likelihood constraint $L_n(\cdot)$ as a functional of the unobserved conditional $p_{x|y,z}$, because the uncertainty set of the joint $p_{x,y,z}$ is expressed in terms of $p_{x|y,z}$. These steps will enable us to solve the robust optimization problem by searching $p_{x|y,z}$ over the uncertainty set.

**Backdoor adjustment.** Under the backdoor criterion, backdoor adjustment estimates the potential outcomes $\mathbb{E}[Y(z)], z \in \{0, 1\}$ by $\mathbb{E}[Y(z)] = \int \mathbb{E}[Y|Z = z, X = x]P(X = x)dx$. If we had access to the noiseless covariates $X$, we estimate the ATE by positing a parametric model $\mathbb{E}[Y|Z = z, X = x] = g(x, z; \theta)$ and calculating $\hat{\tau} = \mathbb{E}[g(X, Z = 1; \theta) - g(X, Z = 0; \theta)]$.

Next we move from noiseless covariates to noisy ones. Given each feasible $p_{x|y,z}$, we inherit the identification formula as the optimization objective $Q(p_{\hat{\theta}}(x, y, z)) = \mathbb{E}_{p_{x|z=1}}[g(X, Z = 1; \theta)] - \mathbb{E}_{p_{x|z=0}}[g(X, Z = 0; \theta)]$, where $p_{x|z} = \int p_{x|y,z} \times p_{y|z}\, dy$. Then the constraint is that $\theta$ maximizes the expected log-likelihood given the dataset: $L_n(\theta\,;p_{x,y,z}) = \mathbb{E}_{Y,Z}[\mathbb{E}_{p_{x|y,z}}[\ell(f(X, Z; \theta), Y)]]$.

**Inverse propensity weighting (IPW).** The application to the IPW method share a similar spirit as backdoor adjustment except that IPW estimates the potential outcome $Y(z), z \in \{0, 1\}$ with a different estimator $\mathbb{E}[Y(z)] = \mathbb{E}[\frac{YZ}{P(Z=z|X)}]$. We estimate the ATE by positing a parametric model on the propensity score, $Z|X \sim \text{Bern}(f(\theta, X))$. Thus for each feasible $p_{x|y,z}$, we can estimate the ATE by $Q(p_{\hat{\theta}}(x, y, z)) = \mathbb{E}_{p_{y,z}}\mathbb{E}_{p_{x|y,z}}[\frac{YZ}{f(X;\theta)} - \frac{Y(1-Z)}{1-f(X;\theta)}]$, which is also the objective of the robust optimization problem. Then the constraint of this problem is that $\theta$ maximizes the likelihood of $p_{z|x}$, i.e. $L_n(\theta \,;\, p_{x,y,z}) = \mathbb{E}_z[\mathbb{E}_{p_{x|z}}[\text{Bern}(Z \,;\, f(X;\theta))]]$ with $p_{x|z} = \int p_{x|y,z} \times p_{y|z} \, dy$.

**Frontdoor adjustment.** Frontdoor adjustment is different from the backdoor adjustment and IPW in that the covariates $X$ serve as mediators between the treatment $Z$ and outcome $Y$. Frontdoor adjustment gives the following estimator for potential outcomes, $\mathbb{E}[Y(z)] = \mathbb{E}_{X \sim P(X|Z=z)}[\sum_{z'=0,1} \mathbb{E}[Y|X, Z = z']P(Z = z')]$. Similar to backdoor adjustment, we can parameterize $\mathbb{E}[Y|X = x, Z = z] = g(x, z\,;\, \theta)$. Thus the ATE identification functional is $Q(p_{\hat{\theta}}(x, y, z)) = \mathbb{E}_{X \sim P(X|Z=1)}[\sum_{z'=0,1} g(X, z';\theta)P(Z = z')] - \mathbb{E}_{X \sim P(X|Z=0)}[\sum_{z'=0,1} g(X, z';\theta)P(Z = z')]$. As we use the same parametric model as in backdoor adjustment, the constraint of the robust optimization problem of the frontdoor adjustment is the same as that of the backdoor adjustment, where $L_n(\theta \,;\, p_{x,y,z}) = \mathbb{E}_{Y,Z}[\mathbb{E}_{p_{x|y,z}}[\ell(f(X, Z; \theta), Y)]]$.

## 5. Experiments

We empirically evaluate the performance of the partial identification for noisy covariates via robust optimization (abbreviated as RCI) on a variety of simulated and real datasets. For each dataset, we synthetically generate noisy versions of it with different noise levels. For each noise level, we compute the noise strength $\gamma_z$ as the TV distance in Eq. 2, which is a parameter of RCI to estimate ATE. We study the performance of RCI applied to a variety of standard causal estimators, including the backdoor adjustment estimator, the IPW estimator and the frondoor adjustment, comparing them with a naive approach that employs the corresponding estimator directly applied to the noisy data. We find that RCI provides partial identification intervals with improved coverage properties than existing approaches, including the Causal Effect Variational Autoencoder (CEVAE) (Louizos et al., 2017), while being not overly conservative (e.g. Fig. 1). We provide the details of the datasets, evaluation procedures and results in sequel. Further data and training details are in Appendix C.

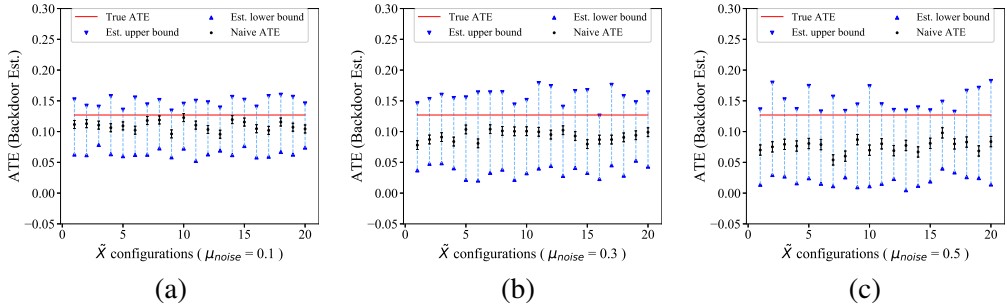

Figure 1: Partial identification of ATE with backdoor adjustment estimators on synthetic dataset with binary outcome. The three noise levels have random Gaussian noise with mean = 0.1/0.3/0.5 and standard deviation = 0.5/0.5/1. In all plots, we compare RCI (this work) to the naive approach. The error bars indicate 95% confidence interval of the naive ATE estimation over twenty trials. *Intervals covering the true ATE is better.*

## 5.1. Synthetic data

**Backdoor adjustment and IPW.** To evaluate the performance of the robust approach with the backdoor adjustment and IPW estimators, we synthetically generate two datasets with $X$ as the confounder. To demonstrate the variability of the estimated ATE intervals, we first synthetically generate a dataset with binary outcomes according to a logistic model. We further consider another synthetic dataset with a more complicated nonlinear outcome model and continuous outcomes, using the Kang and Schafer example (Kang et al., 2007), which consists of four unobserved covariates $U_i \overset{iid}{\sim} N(0, I_4)$, $i = 1, ..., n$. The full data generation details are included in Appendix C.1.

Table 1: Coverage probabilities for the partial identification interval via robust optimization with the backdoor adjustment and IPW estimators, and the naive approach and CEVAE (averaged over 100 trials, with standard error). (*higher* is better).

|         |               |               | RCI           |               |
|---------|---------------|---------------|---------------|---------------|
| Noise   | Naive         | CEVAE         | Backdoor Adj. | IPW           |
| level 1 | $0.10 \pm 0.09$ | $0.46 \pm 0.17$ | $\mathbf{1.00 \pm 0.00}$ | $\mathbf{0.95 \pm 0.02}$ |
| level 2 | $0.00 \pm 0.00$ | $0.41 \pm 0.05$ | $\mathbf{0.98 \pm 0.01}$ | $\mathbf{0.93 \pm 0.03}$ |
| level 3 | $0.00 \pm 0.00$ | $0.42 \pm 0.09$ | $\mathbf{0.97 \pm 0.02}$ | $\mathbf{0.91 \pm 0.03}$ |
| level 4 | $0.00 \pm 0.00$ | $0.33 \pm 0.03$ | $\mathbf{0.95 \pm 0.02}$ | $\mathbf{0.87 \pm 0.03}$ |
| level 5 | $0.00 \pm 0.00$ | $0.32 \pm 0.14$ | $\mathbf{0.93 \pm 0.03}$ | $\mathbf{0.85 \pm 0.04}$ |

**Frontdoor adjustment.** For front adjustment, we synthetically generate two datasets, with $X$ as the mediators. First, we generate a dataset with binary outcomes using a logistic model with a single mediator. We then generate another more complicated using a similar data generation as in Jung et al. (2020). This data generation is more complicated with multiple mediators.

**Noisy data generation.** Given the true covariates, we generate noisy covariates by synthetically adding a small amount of random noise. The ground truth covariates enables us to estimate the true ATE. For each selected example, we perturb it by adding a noise drawn from a Gaussian distribution to each dimension. We then evaluate the performance of the different algorithms ranging from small to large amounts of noise. The full generation details are in C.1.

**Evaluation and results.** To demonstrate the variability of the estimated ATE intervals, we plot the ATE intervals obtained by RCI and the true ATE. As a comparison, we also show the results of the naive ATE estimator, which estimates ATE directly using the noisy examples. The true ATE is calculated using the corresponding adjustment method and the noiseless covariates. We generate 2000 samples for each adjustment method, and generate 20 configurations of the noisy covariates for each noise level.

Figure 1 shows the performance of different algorithms using the backdoor adjustment method. The results with frontdoor adjustment method are similar and included in Appendix C.5. We observe that under the two outcome models, RCI can provide ATE intervals with a high coverage on the true ATE. However, the naive approach is very sensitive to the noise even for the low noise levels, and gives estimations that deviate from the true ATE as the noise level increases. Moreover, the RCI intervals are not overly conservative or trivial: they cover the true ATE without much overshooting. We compute the true ATE coverage probability with the more complicated Kang and

Table 2: Coverage probabilities for the robust optimization approach with frontdoor adjustment, and the naive approach. (a) shows results with simulation data contains multiple mediators (mean and standard errors are averaged over 100 trials.). (b) shows results with the IHDP dataset (mean and standard errors are averaged over 50 trials.) (*higher* is better).

(a)

| Noise | Naive | RCI (Frontdoor) |
|---------|-----------------|------------------|
| level 1 | $0.30 \pm 0.15$ | $\mathbf{0.92 \pm 0.03}$ |
| level 2 | $0.10 \pm 0.09$ | $\mathbf{0.87 \pm 0.03}$ |
| level 3 | $0.00 \pm 0.00$ | $\mathbf{0.84 \pm 0.04}$ |
| level 4 | $0.00 \pm 0.00$ | $\mathbf{0.82 \pm 0.04}$ |
| level 5 | $0.00 \pm 0.00$ | $\mathbf{0.81 \pm 0.04}$ |

(b)

| Noise | Naive | RCI (Frontdoor) |
|---------|-----------------|------------------|
| level 1 | $0.30 \pm 0.15$ | $\mathbf{0.98 \pm 0.02}$ |
| level 2 | $0.10 \pm 0.09$ | $\mathbf{0.96 \pm 0.03}$ |
| level 3 | $0.00 \pm 0.00$ | $\mathbf{0.94 \pm 0.04}$ |
| level 4 | $0.00 \pm 0.00$ | $\mathbf{0.94 \pm 0.04}$ |
| level 5 | $0.00 \pm 0.00$ | $\mathbf{0.92 \pm 0.04}$ |

Schafer example (used for backdoor adjustment and IPW), and the second simulated dataset with multiple mediators for the frontdoor adjustment method. We also compare with CEVAE (Louizos et al., 2017), which identifies ATE via back-door adjustment and models the noised covariates as the proxy variables. A success cover means that the true ATE is contained in the estimated ATE interval by RCI, or by the 95% confidence intervals of naive ATE. For CEVAE, at a specific noise level, we collect its ATE estimates over multiple datasets with noisy covariates. A success cover means the true ATE is within the range of estimates from the noisy datasets. We generate ten random noise-less datasets of true covariates with size 2000. For each noiseless dataset, we further generate ten equal-sized datasets with noisy covariates by drawing fresh noise samples. Therefore, the coverage probabilities are calculated over 100 pairs of true and noisy datasets. Table 1 and Table 2 (*left*) show the coverage probabilities using the three adjustment methods. We see that RCI is able to maintain a much higher coverage probability as the noise level increases.

## 5.2. Real data case studies

We further test the robust approach RCI on two case studies with real covariates, including an ACIC dataset and an IHDP dataset. Both datasets have been used for benchmarking various causal inference algorithms (Shalit et al., 2017; Shi et al., 2019; Gupta et al., 2020).

**Case study 1: ACIC dataset.** We first use a dataset constructed for the Atlantic Causal Inference Conference (ACIC) 2019 Data Challenge based on the "spambase" dataset for spam email detection from UCI (Gruber et al., 2019; Dua and Graff, 2017). This dataset consists of emails with an outcome of interest $Y$ being whether or not the email was marked as spam by a user. The treatment $Z$ represents whether or not the email contains more than a given threshold of capital letters, where this threshold is computed by a mean over the original dataset. There are 22 continuous covariates $X$ which are word frequencies given as percentages between 0 and 100. We generate our dataset directly using ACIC's data generating process, with a size of 2000 examples. Given the true covariates, we further generate noisy covariates by synthetically adding a small amount of noise at random, using a similar procedure as for the synthetic data. Specifically, we generate five levels of Gaussian noise with mean = 0.1/0.2/0.3/0.4/0.5 and standard deviations at 0.5/0.5/1/1/1.

**Case study 2: IHDP dataset.** For a second case study, we use a benchmark dataset introduced by Hill (2011), which is constructed from data obtained from the Infant Health and Development

Table 3: Coverage probabilities for the robust optimization approach with the backdoor adjustment and IPW estimators, and the naive approach, using the ACIC dataset. (The results are averaged over 50 trials). (*higher* is better).

| Noise | Naive | CEVAE | RCI | |
| --- | --- | --- | --- | --- |
| | | | Backdoor Adj. | IPW |
| level 1 | $0.02 \pm 0.02$ | $0.81 \pm 0.07$ | $\mathbf{1.00 \pm 0.00}$ | $\mathbf{1.00 \pm 0.00}$ |
| level 2 | $0.00 \pm 0.00$ | $0.73 \pm 0.05$ | $\mathbf{0.98 \pm 0.02}$ | $\mathbf{0.98 \pm 0.02}$ |
| level 3 | $0.00 \pm 0.00$ | $0.75 \pm 0.10$ | $\mathbf{0.94 \pm 0.03}$ | $\mathbf{0.92 \pm 0.04}$ |
| level 4 | $0.00 \pm 0.00$ | $0.64 \pm 0.02$ | $\mathbf{0.94 \pm 0.03}$ | $\mathbf{0.90 \pm 0.04}$ |
| level 5 | $0.00 \pm 0.00$ | $0.64 \pm 0.03$ | $\mathbf{0.90 \pm 0.04}$ | $\mathbf{0.90 \pm 0.04}$ |

Program (IHDP). This dataset is based on a randomized experiment to measure the effect of home visits from a specialist on future test scores of children. The confounders $U$ correspond to collected measurements of the children and their mothers used during a randomized experiment that studied the effect of home visits by specialists on future cognitive test scores. We use samples from the NPCI package (Dorie, 2016), which converted the randomized data to an observational study by removing a biased subset of the treated group. The final dataset contains 747 samples with 25 covariates. We then simulate the mediator and the outcome using a procedure similar to Hill (2011); Gupta et al. (2020). We generated the noisy covariates using the same five noise levels as the ACIC dataset. The full generation details are in C.4.

**Evaluation results.** We evaluated the naive approach, the RCI approach with the backdoor adjustment and IPW adjustment methods on the ACIC dataset. We also evaluated the naive approach and RCI with the frontdoor adjustment method on the IHDP dataset. Table 3 and Table 2(*right*) show the coverage probabilities using these three adjustment methods. For both case studies, RCI is able to maintain a much higher coverage probability as the noise level increases, while the naive approach's estimates turn out to be very sensitive to the noise and have low coverage probabilities. Frontdoor adjustment method is able to achieve a higher coverage probability comparing to the synthetic data. This could be due to the fact that, in this data generation model, the outcome is linearly correlated with the mediator. As we also used a linear parameterized model, there is no model specification.

## 6. Conclusion

This paper develops an approach to partial identification for noisy covariates via robust optimization. We show that partial identification can be formulated as a robust optimization problem, which enables bounds on causal effects for parametric causal models. We then derive a variant of the projected gradient algorithm to efficiently solve the robust optimization problem and compute partial identification bounds on the causal effect of interest. We illustrate the wide applicability of our approach on a variety of causal adjustment methods, including the backdoor adjustment, inverse propensity weighting and the frontdoor adjustment. Numerical results across synthetic and real-world data show that this approach can effectively compute bounds for ATE with higher coverage than previous methods without being overly conservative.

## Acknowledgments

The authors would like to thank Peng Ding for extensive discussions and helpful suggestions that significantly improved the paper. The authors also thank Peter Bickel, Avi Feller, Sam Pimentel, Vira Semenova, and Yan Shuo Tan for helpful feedback on early versions of the paper. This work was supported in part by the Mathematical Data Science program of the Office of Naval Research under grant number N00014-18-1-2764. WG acknowledges support from a Google PhD fellowship; MY acknowledges support from the Irving Institute for Cancer Dynamics.

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
