# OpenReview forum: "Partial Identification with Noisy Covariates: A Robust Optimization Approach"
_cclear.cc/CLeaR/2022/Conference — CLeaR 2022 Poster_

### Official Review · Reviewer_r5az · 2021-11-19

**Confidence:** 4
**Overall Score:** 6

**Main Review:**

This paper studies an interesting and important problem in causal inference, namely partial identification analysis of treatment effects when covariates cannot be measured accurately. The authors propose a distributionally robust optimization approach where the uncertainty set consists of all possible conditional distributions of noise-free covariate in a total-variation ball around the conditional distribution of noisy covariates. This approach is very sensible. The writing is overall concise but clear.

The following are my major questions/comments about this paper:
- The Lagrangian formulation in equation (8) does not look quite straightforward. Is this formulation used to replace the optimality constraint in equation (7)? But equation (8) still has $p_{\hat\theta}$; then what's the definition of $\hat\theta$ after dropping the constraint in eq. (7)? Or is eq (7) still implicitly maintained in eq. (8)? Plus, this constraint in eq. (7) means that the whole problem in (6)(7) is a bilevel problem, which is usually difficult to solve. I am not sure whether the simple lagrangian formulation is enough to deal with it. I suggest the authors justify the formulation more rigorously.
- In section 4, the likelihood functions all involve integration w.r.t conditional distribution of the noise-free covariate. This appears challenging. How did the authors deal with this?
- In most of the experiments, the authors only showed the coverage probabilities but not width of the proposed intervals. So there are only quite limited evidences for "without being overly conservative".

Minor comments:
- On page 8 and 9, what's the meaning of the function $f(X, Z;\theta)$ at the end of paragraphs named **Backdoor adjustment** and **Frontdoor adjustment**? Are they typos?


**Summary:**

This paper proposes a robust optimization approach to learn the partial identification bounds of ATE when covariates have measurement errors.

---

### Official Review · Reviewer_yzHU · 2021-11-19

**Confidence:** 2
**Overall Score:** 6

**Main Review:**

In my view, this paper provides meaningful bounds for a plausible sensitivity assumption. I think that achievement on its own is acceptable.

I think the paper could go from acceptable to great by doing one or more of the following:

(1) Making a better case that the TV bound assumption is either intuitive or especially tractable

(2) Explaining where the identification comes from and which assumptions are made only for convenience

(3) Making the case that the resulting bounds are easy to compute.

All may hold, but the paper does not do a convincing job of arguing that they do at the moment

I do appreciate the argument that TV bounds are a weakened version of various existing models (Section 2). Still, tight bounds for TV bounds may not be tight for the strengthened assumptions, so it would be good to have more intuition for how TV works and what TV means to appreciate those bounds for their own sake.

My small worry is that I am not sure the experiments are that meaningful. If I read correctly, their experiments ask whether we cover the true ATE rather than the identified set. Once we believe in noisy covariates, I do not see why the ATE used to generate the noisy covariates is meaningful. Though perhaps I am missing something.

My big worry, which may reflect a misunderstanding on my part and which perhaps could be avoided by deleting a few paragraphs, is that the application to sensitivity analysis on page 8 is too blasé. If Y is unbounded, then I expect a bound on $TV( p_{e(x_{inc})}, p_{e(x_{full})})$ to have zero identification power. Consider the following example:

Suppose we observe no covariates and that $E[Z] = 1/2$, so that $e(x_{inc}) = 1/2$ constantly. Suppose also the distribution of $Y|Z$ is unbounded and continuous. Construct a dataset as follows. Draw some $V \sim Unif(0, 1)$ and $\hat{Z} \sim Bern(0.5)$ iid. Set $Z = \hat{Z}$ and $Y(\hat{Z}) = Q_{V}(Y | Z = \hat{Z})$. If $V \in [\frac{\gamma}{2}, 1-\frac{\gamma}{2}]$, set $Y(1-\hat{Z}) = Q_{V}(Y | Z = 1-\hat{Z})$. If $V \not{\in} [\frac{\gamma}{2}, 1-\frac{\gamma}{2}]$, sample $Y(1-\hat{Z})$ from some density over the support of $Y | Z=1-\hat{Z}, F_{Y|Z=1-\hat{Z}}(Y) \not{\in} [\frac{\gamma}{2}, 1-\frac{\gamma}{2}]$. Finally, set $x_{full} = (Y(1), Y(0))$, so that $e(x_{full}) = E[Z | Y(1), Y(0)]$

By construction, $V \in (\frac{\gamma}{2}, 1-\frac{\gamma}{2})$ if and only if $F_{Y | Z = z}(Y(z)) \in (\frac{\gamma}{2}, 1-\frac{\gamma}{2})$ for both $z$, so for such $Y(1),Y(0)$ we have $E[Z | Y(1), Y(0)] = 1/2$. If we write $\pi$ for the distribution over $\left(e(x_{inc}), e(x_{full}), V\right)$, then $\pi$ marginalizes to the right propensities and $$E_{\pi} \left[ \mathbb{I} \left[ e(x_{inc}) \neq e(x_{full}) \right] \right] \leq E_{\pi} \left[ \mathbb{I} \left[ V \not{\in} [\frac{\gamma}{2}, 1-\frac{\gamma}{2}] \right] \right] = \gamma$$
Therefore the TV distance between propensity distributions would, if I understand correctly, always be $\leq \gamma$. On the other hand, we can sample $Y(1), Y(0)$ among the most extreme outcomes arbitrarily (so long as we have a propensity). We can have a non-trivial fraction $1/(4 \gamma)$ of the data sample $Y(1)$ from its arbitrarily-large most extreme $1/M$ fraction of the data for an arbitrarily-large $M$. In other words, the bounds would seem to be infinite.

I would like to understand whether the TV bounds are finite in my particular example (so that the discussion of TV bounds should be clearer to avoid others erring in the same direction), are infinite usually with unbounded outcomes (so that the text needs a meaningful caveat), or my example can be distinguished from the standard case (so that the discussion of sensitivity analysis should be changed/removed and the paper should be clearer on the source of its identifying power relative to my example). I would like to know where I am going wrong or, if I am not going wrong, either why this method is applicable to sensitivity analysis or why the other applications are distinguishable.


**Summary:**

The paper studies partial identification under TV distance bounds on noisy covariates. They propose a robust optimization approach and provide methods for inference.

---

### Official Review · Reviewer_SQvi · 2021-11-20

**Confidence:** 4
**Overall Score:** 6

**Main Review:**


Originality:

The authors of the paper seem not to recognize that this problem can already be
exactly solved in the discrete case by existing methods. The authors write of
these works:

> Our work is complementary to this work; we focus on noisy covariates but can
handle certain settings with continuous variables, without relying on additional
compliance assumptions.

Moving from discrete variables, whose distributions have a finite number of
parameters, to parametric models of continuous variables, whose distributions
again have a finite number of parameters, is not an especially original
innovation. That said, there is clearly value in pointing out that these
constrainted optimization methods can be applied whenever there are a finite
number of parameters, as in parametric models of continuous variables.

Regarding the originality of the optimization procedure, the authors write in
the introduction

> we develop an algorithm for partial identification using robust optimization.

However when the algorithm is introduced, they write

> To solve Eq. 11, we use a projected gradient descent ascent (GDA) algorithm,
  which is a simplified version of the algorithm introduced by Namkoong and
  Duchi (2016) for solving general classes of DRO problems.

It is not clear that this qualifies as the development of an algorithm! I would
prefer to see the contribution of the paper framed as developing a formulation
of the partial identification problem that can be solved by existing algorithms
from the robust optimization literature.

Significance:

The approach presented in this paper has pros and cons relative to the analogous
approaches in other works. On the pros side, this approach only bounds the piece
of the identifying functional that the measurement error blocks identification
of. As a result, the constrained optimization problem has less work to do, in
that more of the structure of the problem is encoded.

On the cons side, this paper specifies but a single measurement error
assumption, which may or may not be well suited to the task at hand. I
appreciate the efforts in Section 2 to show that the TV distance can be applied
in many cases, but clearly some flexibility is lost. This method also applies in many
fewer settings than more general approaches (e.g. Duarte et al 2021).

It seems to me that the main contribution of this paper is in suggesting the use
of algorithms from the DRO literature to solve partial identification problems.
What I did not understand from the paper was whether there is any benefit to the
DRO approach over existing constrained optimization approaches. I imagine there
might be, as other constrained optimization approaches become computationally
intractable quickly, so certainly alternative formulations of these problems are
of interest. In addition, the idea of only bounding factors of the district
factorization that are not identified can help with computational tractability
as well.

Regarding the experiments, I think the comparisons with CEVAE and the "naive"
approach not very interesting. CEVAE does not provide any guarantees, and the
"naive" approach will do badly for the same reason that estimates under
misspecified models are always worse than estimates under correctly specified
models. That said, I also do not think these comparisons are necessary. This
approach, like other constrained optimization approaches, produces bounds that
are sharp under the assumptions. If sharpness is theoretically guaranteed, there
is not much need for numerical examples. I think more interesting would be
experiments that examine how much tightness is lost by using the TV measurement
error assumption relative to other measurement error assumptions. The point of
the end of section 2 is that you can always find a TV ball around the
distribution that includes all distributions permitted by other assumptions, but
it is important to point out as well that the opposite does not hold, i.e. the
TV assumption will in most cases be weaker than the assumption it replaces.

Overall, I believe this work is worth publishing for introducing the partial
identification literature to a potentially useful toolkit from the DRO
literature. However, I think it would be much improved if it engaged more deeply
with existing approaches to partial identification based on constrained
optimization, and was ambitious enough to attack the more general problems
addressed by those methods. I also think it would be improved by discussion of
computational feasibility, especially in relation to existing approaches. How
long do the problems presented in the text take to run? How does runtime scale
in the number of parameters used to model the relevant distributions?

Technical Quality and Clarity:

The paper is correct and clear. There are small typos throughout, and there are
several arXiv citations to papers that are published (including the Gieger &
Meek 1998 paper, which was published 15 years before someone uploaded papers
from that conference to arXiv).


**Summary:**

The method of this paper provides bounds on the ATE in the setting where the ATE of Z on Y is identified if X is observed, but we have only a noisy measure of X. The paper is largely well-presented and well written, and I appreciate that the code and data are made available in the supplementary materials.

---

### Decision · Program_Chairs · 2022-01-12

**Decision:**

Accept (Poster)

**Comment:**

In this paper, the authors bring the tools of robust identification literature to bear on the problem of partial identification of causal parameters.

The reviewers appreciated this connection, but had concerns about experiments, and proper engagement with existing approaches to partial identification developed in the causal inference community, particular for discrete data models.

However, the paper was received positively by reviewers, and subsequent discussion addressed their concerns.

As a result, the paper is suitable for publication in CLEAR.